behaviour/ecology/environmental science

thermal preference, set-point, thermoregulation, oxygen sensing, hydrogen sulfide

**Author for correspondence:**
Glenn J. Tattersall
e-mail: gtattersall@brocku.ca

# Hydrogen sulfide exposure reduces thermal set point in zebrafish

Dimitri A. Skandalis, Cheryl D. Dobell, Joshua C. Shaw and Glenn J. Tattersall

Department of Biological Sciences, Brock University, St Catharines, 500 Glenridge Avenue, St Catharines, Ontario, Canada L2S 3A1

DAS, 0000-0002-5126-9582; GJT, 0000-0002-6591-6760

Behavioural flexibility allows ectotherms to exploit the environment to govern their metabolic physiology, including in response to environmental stress. Hydrogen sulfide ($H_2S$) is a widespread environmental toxin that can lethally inhibit metabolism. However, $H_2S$ can also alter behaviour and physiology, including a hypothesized induction of hibernation-like states characterized by downward shifts of the innate thermal set point (anapyrexia). Support for this hypothesis has proved controversial because it is difficult to isolate active and passive components of thermoregulation, especially in animals with high resting metabolic heat production. Here, we directly test this hypothesis by leveraging the natural behavioural thermoregulatory drive of fish to move between environments of different temperatures in accordance with their current physiological state and thermal preference. We observed a decrease in adult zebrafish (*Danio rerio*) preferred body temperature with exposure to 0.02% $H_2S$, which we interpret as a shift in the thermal set point. Individuals exhibited consistent differences in shuttling behaviour and preferred temperatures, which were reduced by a constant temperature magnitude during $H_2S$ exposure. Seeking lower temperatures alleviated $H_2S$-induced metabolic stress, as measured by reduced rates of aquatic surface respiration. Our findings highlight the interactions between individual variation and sublethal impacts of environmental toxins on behaviour.

## 1. Introduction

Environmental toxicants may act through myriad pathways, including hijacking the body's own signalling pathways. Hydrogen sulfide ($H_2S$) is a widespread aquatic toxicant that is also an important endogenous gasotransmitter, occurring naturally through anoxic decomposition (e.g. salt marshes and mangrove swamps) or due to anthropogenic activities (e.g. sewage treatment

and aquaculture farming) [1,2]. Exogenous $H_2S$ inhibits aerobic respiration [3] and, together with low oxygen (hypoxia), contributes to large fish kills [1,2,4]. However, $H_2S$ is not exclusively toxic and has endogenous roles including the physiological response to hypoxia and regulation of synaptic activity, cognitive function, inflammation and oxygen sensing [3,5–7]. It has been proposed that application of exogenous $H_2S$ in combination with low temperatures induces a drop in body temperature through entry into a hypometabolic hibernation-like state in mice [8]. It is unclear if this is an effect of $H_2S$ alone or aggravation of a conserved environmental hypoxia response [9,10]. These studies have been performed in small mammals within their thermoneutral zone, where thermogenesis and dissipation are normally balanced; metabolic poisoning by exogenous $H_2S$ might impair resting heat production rather than stimulate a controlled depression of the set point. By contrast, ectotherms like fish behaviourally defend their thermal preference, enabling direct assessment of body temperature regulation. Whereas in most terrestrial animals exogenous $H_2S$ is applied to study the gasotransmitter's endogenous functions [8,9], exogenous $H_2S$ is ecologically relevant in aquatic habitats [11–14]. We exploit this physiology as a direct test of the hypothesis that $H_2S$ drives changes in thermal preferences, which is significant for the ecology and behaviour of this major taxon.

Fish can detect water temperature changes of 0.05°C or less [15] and active fish, like zebrafish (*Danio rerio*), tend to move toward their preferred temperatures [16]. Preferred temperatures vary within and among individuals, depending on factors like growth, health, acclimation, metabolic state and social cues [17–23]. Numerous fish species select colder temperatures (i.e. behavioural anapyrexia) in hypoxia than in normoxic conditions [24–26], presumably due to enhanced haemoglobin oxygen-binding capacity and reduced metabolic demand of tissue at low temperatures, which balance oxygen supply and demand [24]. Exposure of fish to $H_2S$ shares many physiological similarities with hypoxia [1,26], possibly because $H_2S$ metabolism functions as an endogenous oxygen sensor [14,26] to reversibly bind mitochondrial cytochrome oxidase and compete with oxygen [3]. Here, we examine how zebrafish thermal preferences are altered with exposure to $H_2S$ in normoxic conditions. We tested the hypothesis that $H_2S$ triggers a reduction in individual thermal set point, pointing to sublethal effects of $H_2S$ on physiology and behaviour.

## 2. Materials and methods

Zebrafish (*Danio rerio*) from a local supplier were housed in 40 l aquarium tanks at 27°C and pH 7.6–7.8 (Seachem™ Acid Buffer), on a 12 L : 12 D cycle with once-daily feedings (Tetra Flakes™). Fish were housed at least 24 days and habituated to walls lined with white contact paper (required for the automatic camera tracking software) prior to experiments, to mitigate the stress of a change in the visual environment. Moreover, dark walls may facilitate stress responses that affect subsequent behavioural trials [27]. No individuals were obviously ill, judged by pre-test condition and robust escape responses. Total sample sizes in each condition were $n = 20$ for 0% $H_2S$ and $n = 17$ for 0.02% $H_2S$.

Thermal preferences were tested in a two-chamber dynamic shuttlebox (see electronic supplementary material, figure S1) described previously [28] by automatically tracking body position ($x,y$; $x = 0$ at midline, $+x$ to the right), swim velocity and chamber temperatures (1 Hz sampling, ICFish v. 2.1, Brock University Electronics; see [28]). In response to a separate series of trials at a constant temperature, fish quickly habituated to the chamber and shuttling rate decreased to a constant level within one hour (see electronic supplementary material, figure S2). Hydrogen sulfide was bubbled through side chambers inaccessible to fish, but in fluid contact with the main chamber, allowing continuous gas equilibrium without disturbing the fish. Air and 0.2% $H_2S$ (Praxair certified) were first mixed volumetrically to achieve the appropriate $H_2S$ concentration (0% or 0.02% $H_2S$) using two calibrated flow meters (Omega rotameters) to achieve a total constant flow of 5000 ml min$^{-1}$ (0.07% $H_2S$ elicited severe distress, not shown). Bubbling mixed gases avoids difficulties in determining $H_2S$ concentration from dissolved NaS salts [6,26], and balancing $H_2S$ with air (rather than nitrogen) guarantees normoxia (20.88% $O_2$). Gas dissolution equilibrated for 30 min, and pH was buffered within the range of 7.6 to 7.8. Pilot experiments, measuring $H_2S$ using an $H_2S$ electrode, demonstrated constant concentrations throughout an 8 h period of gas bubbling. Gas was also flowed under a clear Plexiglas cover (5 mm above the water surface) to maintain constant air space gas pressures and minimize condensation. Pilot experiments revealed robust thermoregulatory behaviour when fish first learned to associate each chamber with a constant temperature difference. Fish were introduced to the left chamber (figure 1, set to 1.5°C below housing temperature) and habituated 1 h with a constant 3°C difference between chambers. Ramping then commenced for 2 h, triggered when the fish entered the left (cooling, −0.5°C min$^{-1}$) or right

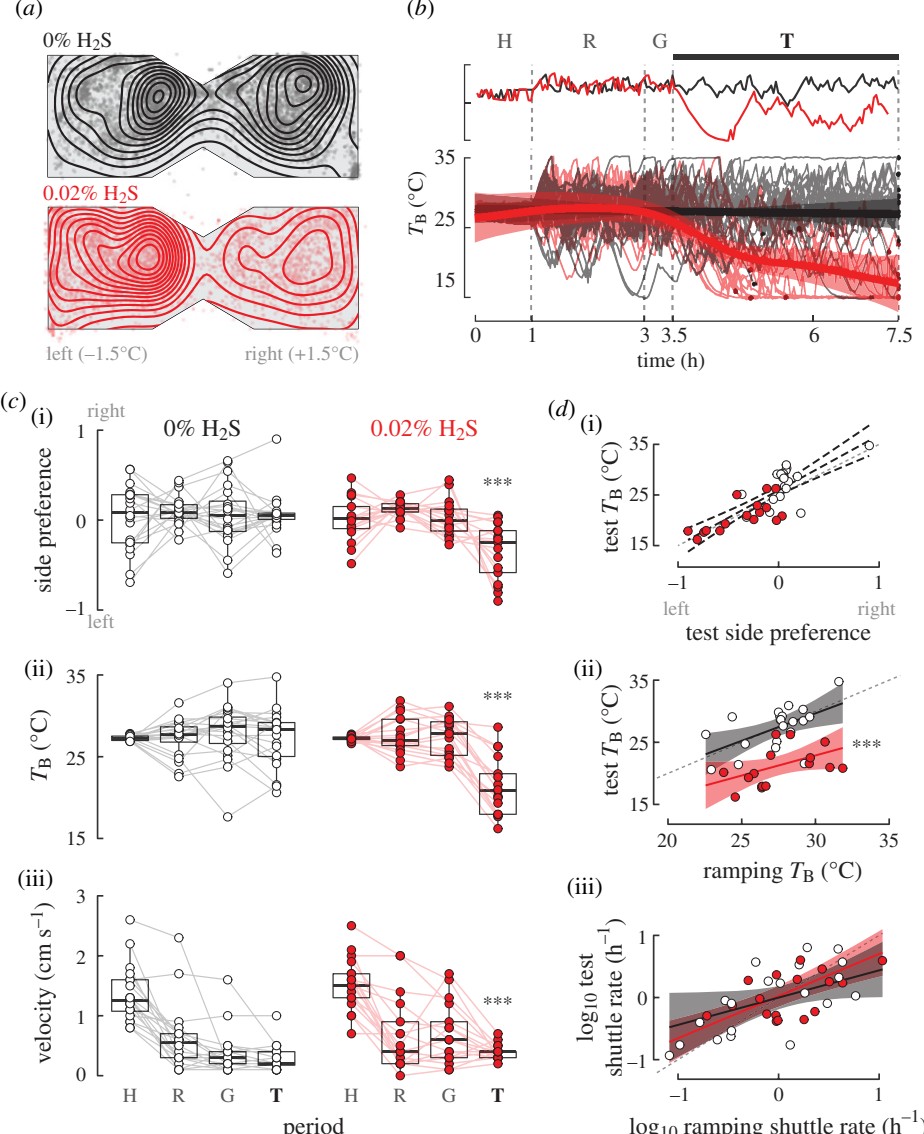

**Figure 1.** Hydrogen sulfide ($H_2S$) exposure drives zebrafish behavioural anapyrexia. (a) Adult zebrafish actively defend body temperature ($T_B$) by shuttling between chambers of 3°C difference. Habituation (H), ramping (R), gas (G) and testing (T) phases are depicted. Average fish position reveals cold preferences (left) with exogenous 0.02% $H_2S$ (red). Bounding boxes are estimated for visualization only. (b) Body temperature (right, thick lines) is constant at 0% $H_2S$ but decreases rapidly in 0.02% $H_2S$. *Top*: representative individual traces; *Bottom*: thin lines are individual traces, thick lines are population average. Termination times for individuals are shown by a solid dot. (c) Progression of period-averaged responses. Side preference (i), body temperature (ii), and swim velocity (iii) significantly differed in 0.02% $H_2S$ (***: $p < 0.001$). (d) (i) Interindividual variation in $T_B$ across treatments was correlated with time spent in each chamber ($p = 0.001$). All individuals must lie on this trend by construction, so treatment effects were excluded (distinguished by dashed CI band). (ii) Preferred $T_B$ during ramping and test phases were correlated with slope near unity, but with a significant intercept shift due to 0.02% $H_2S$, pointing to a change in thermal set point. (iii) Shuttle rate was correlated, with slope less than unity, suggesting fish learn to defend $T_B$ even while expending less effort (fewer shuttles). Sample sizes: $n$ 0% $H_2S = 20$; $n$ 0.02% $H_2S = 17$.

(warming, +0.5°C min⁻¹) sides, within limits set to 15 and 35°C. These temperatures are within 5–10°C of *D. rerio*'s thermal tolerances [20]. Following 30 min gas equilibration, behaviour was recorded for 4 h (test phase). Fish that exhibited distress (e.g. loss of equilibrium, excess time at surface) were pre-emptively removed from the experiment. Fish $x$ position was used to calculate shuttling rate (frequency of crossing $x = 0$, min⁻¹) and side preference $2 \cdot (0.5 - \text{Time}_{x<0}/\text{Time}_{\text{total}})$. Thermal inertia of small fish is minimal compared to water temperature [19,29,30], so we calculated body temperature ($T_B$) by averaging the current chamber temperature with $T_B$ in the previous time step. Lower and upper escape temperatures (LET and UET) were the last recorded $T_B$ prior to a shuttle. In preliminary experiments,

thermoregulatory responses of fish to hypoxia ($2\%$ $O_2$) were examined for consistency with documented behaviours [25]. Fish reduced $T_B$ in hypoxia (one-tailed $t$-test with unequal variances, $t_{11.4} = 2.01$, $p = 0.03$, $\Delta T_B \sim -3.2°C$) and decreased swim speed ($t_{13.9} = 1.85$, $p = 0.04$, $\Delta$ speed $\sim 0.90$; see also [25]). Our design therefore detects behavioural anapyrexia in zebrafish expected from observations of other fish species (e.g. [24]).

Respiratory responses were assessed in two fish simultaneously at a constant temperature, each in one of the chambers and separated by an opaque barrier. Six individuals were exposed to each combination of 0 or $0.02\%$ $H_2S$ and 21 or $28°C$ (approximating thermal preferences from shuttle box experiments) and two individuals to $0\%$ $H_2S$ at $28°C$, for 60 min. The aquatic surface respiration rate was estimated by counting the proportion of video frames in which the fish was at the surface (1 Hz sampling, [25]; analysed in ImageJ v. 1.52).

All variables were analysed in the R language through linear, generalized linear and generalized additive models [31–33]. For visualization, shuttlebox walls were estimated *post hoc* from fish positions, and densities clipped to those borders. We applied generalized additive models (*mgcv* [32]) to model the difference in average $T_B$ over time between $0\%$ (reference spline) and $0.02\%$ (difference spline) $H_2S$. Serial autocorrelation of time series model errors was enforced through a Gaussian process spline basis with AR(1) autocorrelation structure and $\rho = 0.95$. Random variation among individuals was incorporated through first-order random smooths [32]. Generalized additive models were likewise applied to analyse the time course of shuttling rates, which were modelled as zero-inflated Poisson processes (see electronic supplementary material, figure S1). All other variables were quantitatively analysed through linear models. Fish velocity and shuttling rate were log-transformed prior to analysis. To quantify individual variation in responses to $H_2S$, we assessed the relationships of responses during the ramping phase to those during the testing phase. This approach is predicated on the consistency of intra-individual thermal preferences over the course of the experiment, which we justify by calculating repeatabilities (*rptR* [33]) of thermal preferences and behaviour between the ramping and testing phases, within the $0\%$ $H_2S$ group. We report confidence intervals (CI) of effect sizes and associated $p$-values (two-tailed, $\alpha = 0.05$), with full model summaries included in the electronic supplementary material.

The effects of $H_2S$ and temperature on the probability of finding fish at the surface (aquatic surface respiration) in our second series of experiments were modelled with a binomial error distribution using Markov chain Monte Carlo (R package *brms* [31]). We examine the log odds ratio of finding a fish at the surface (success), given the total number of recorded frames (trials; see also [25], analysed in ImageJ v. 1.52). We ran four chains of 10 000 iterations each to convergence ($\hat{R} = 1$, [34]), discarding half as burn-in. Significance was interpreted as posterior parameter $95\%$ credible intervals (CrI) excluding zero. Qualitative differences of side preference were visualized through two-dimensional kernel density estimates with a bandwidth of $50 \times 50$ pixels, unconstrained by shuttlebox boundaries.

## 3. Results

When exposed to $H_2S$, fish distinctly preferred the cold chamber (two-dimensional kernel density plots of position in figure 1; side preference in $0\%$ $H_2S$, CI: -0.18–0.06; in $0.02\%$ $H_2S$, CI: 0.21–0.48). The change in side preference upon exposure to $H_2S$ was rapid (time course in figure 1*b*; GAM statistics in supplementary statistical tables), resulting in significantly reduced $T_B$ (figure 1*cii*; $p < 0.001$; $\Delta T_B$ CI: 3.6–8.4°C), lower ($p = 0.002$; $\Delta$LET CI: 1.6–6.1) and upper ($p = 0.001$; $\Delta$UET CI: 1.7–6.7) escape temperatures. Several fish entered the cold side and stopped shuttling altogether (figure 1*b*), which could mean that metabolic stress drove an escape response in fish that subsequently became trapped on the cold side. However, shuttling rates did not differ overall ($p = 0.60$; CI: -0.23–0.40 $\text{min}^{-1}$), and swim velocity actually increased after the introduction of gas (figure 1*ciii*), including in the testing phase ($p = 0.007$; CI: 0.21–0.68 cm s$^{-1}$). Thus, we do not find evidence for behavioural impairment that may have caused this side preference.

Fish side selection was dependent on variation in individual temperature preference (figure 1*di*), despite acclimation together at $27°C$ for longer than a typical period of 10–12 days (e.g. [20,21]). The consistency of preferred $T_B$ over the experiment duration in $0\%$ $H_2S$ (traces in figure 1*b*) suggested that ramping phase $T_B$ could be used to gauge how temperatures are selected in the testing phase (figure 1*b*). In $0\%$ $H_2S$, a constant preferred temperature is indicated by a regression slope overlapping unity (figure 1*dii*; slope 0.88, CI: 0.24–1.53) and moderate repeatability ($R = 0.54$, CI: 0.13–0.78, $p = 0.006$). The high correlation and conserved preference might be surprising given that fish must learn the paradigm during the ramping phase. The effect of learning instead appears to be in the

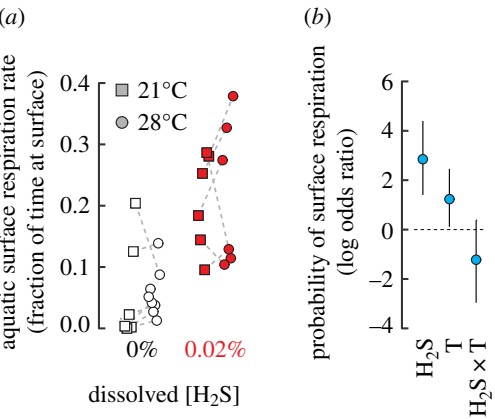

**Figure 2.** Hydrogen sulfide and body temperature drive aquatic surface respiration. (*a*) Adult zebrafish exposed to 0.02% H$_2$S (red) exhibit greater aquatic surface respiration rates (ASR, fraction of time at surface) compared to 0% H$_2$S. (*b*) The probability of surface respiration increases approximately 17-fold with exposure to H$_2$S (from 0% to 0.02%), increases approximately 3.4-fold with increased water temperature (21 to 28°C), but is not significantly affected by the interaction of these parameters (odds ratio: 0.05–1.47).

shuttling rate, which is similarly repeatable (figure 1*d*iii, log shuttle rate $R = 0.51$, CI: 0.08–0.77, $p = 0.01$) but with slope less than unity (CI: 0.38–0.98; no significant effect of treatment, H$_2$S $p = 0.85$; log shuttling rate × H$_2$S: $p = 0.37$). The low slope suggests that fish fine-tune behaviour to maintain preferred $T_B$ with less effort. Given the repeatable $T_B$, we examined how H$_2$S exposure alters individual thermal set point. If H$_2$S causes a reduction in a fish's innate set point, we would expect an intercept difference alone between the ramping and test $T_B$ relationship. Conversely, if H$_2$S causes severe distress and an escape response so that fish try to achieve the lowest possible temperature regardless of their innate set point, we would expect to observe a significant $T_B$ × H$_2$S interaction. The intercept shift was not accompanied by an interaction (figure 1*d*ii; $p < 0.001$; CI $\Delta T_B$: 4.1–8.3°C; $T_B$ × H$_2$S: $p = 0.61$), pointing to a reduction of $T_B$ set point.

During the H$_2$S trials, we observed numerous fish high in the water column, presumably attempting to access what could have been an oxygen-rich surface layer. Our experimental design precluded oxygen or hydrogen sulfide gradients, suggesting that aquatic surface respiration (ASR) was driven by a reflexive respiratory drive rather than detection of greater oxygen or reduced H$_2$S near the surface, and confirming previous records of the behaviour in H$_2$S and hypoxia [25,26]. We therefore tested whether fish held at the mean temperatures of the control and H$_2$S conditions (approximately 21 and 28°C; figure 1*b,c*ii) would exhibit a reduced ASR consistent with alleviation of distress by seeking colder temperatures. The odds of finding fish at the surface increased approximately 17-fold with exposure to 0.02% H$_2$S (figure 2, log odds ratio credible interval, CrI: 1.40–4.49) and approximately 3.5-fold with exposure to higher temperatures (log odds ratio CrI: 0.11–2.46).

## 4. Discussion

A central question in thermoregulatory physiology is the nature of the thermal set point and how it is adjusted [16,18]. Adult zebrafish temperature preferenda spanned 10°C (figure 1) and were consistently reduced approximately 6°C during H$_2$S exposure (figure 1*d*ii) despite greater swim speeds and constant shuttling frequencies (e.g. figure 1*c*). These observations point to the defence of a new thermal set point, similarly to the widely conserved anapyrexic response to hypoxia [10,24,35]. The change in body temperature is consistent with the view that H$_2$S is a key effector of hypoxia sensing in fishes' neuroepithelial cells (NECs, [5,25,26,36]), which contain H$_2$S-producing enzymes [26] that enhance H$_2$S production in response to changes in oxygen. Exogenous H$_2$S greatly increases ventilatory rates and accentuates physiological responses to hypoxia [9], and partially rescues hypoxic ventilatory responses when NECs are inhibited [26]. In mammals, H$_2$S alone or in combination with hypoxia induces anapyrexia [8,9]. Our set-up precluded hypoxia, and so we can conclude that in fish, if not in mammals, H$_2$S induces hypometabolism rather than functioning as a hypometabolic adjuvant [9]. Mammalian carotid bodies and NECs are physiologically similar [10,14,26], so a better

understanding of the cellular roles of $H_2S$ in hypoxia sensing could illuminate potential functions like inducing artificial hibernation [8].

Hydrogen sulfide drives fish to seek alternative environments, including through emersion [1,11] or refuge in habitats such as estuaries [4]. We found that $H_2S$ exposure causes fish to seek colder temperatures, which results in depressed aquatic surface respiration rates (tested temperatures of 21 and 28°C coincide with average 0% $H_2S$ $T_B = 27.3$°C and 0.02% $H_2S$ $T_B = 21.3$°C). Environmental $H_2S$ could therefore impact daily and seasonal habitat selection [1,21] by driving fish to cooler waters and contribute to intraspecific segregation and selection during colonizations of $H_2S$-replete habitats [11,13]. The extent of $H_2S$'s effects would probably depend on the value fish place in maintaining a habitat or defending a territory [22,23,28]: fish with relatively higher temperature preference might be more resistant overall and less likely to abandon a current habitat in favour of searching for alternative environments. Low temperatures in cold refugia can also drive redox reactions that release $H_2S$ from mud [37], mitigating the value of the anapyrexic behavioural response. These thermoregulatory changes can help reduce toxicity by reducing the magnitude of effects despite prolonging duration, and this action may be a central force in the evolution of anapyrexia [38,39]. The ultimate adaptive value of the behaviour will therefore depend on the level of environmental $H_2S$ and the combination of direct physiological and indirect impacts on microfauna and flora [4] that affect habitat suitability. Overall, we find that the capacity of $H_2S$ to alter behavioural thermal preferences in the absence of hypoxia [9] contributes to its complex environmental effects [1,4]. The potency of this effect might reflect its critical role in sensing and responding to oxygen levels, suggesting that environmental hijacking of an endogenous gasotransmitter can profoundly affect animal behaviour.

Ethics. Brock University Animal Care Committee approved experimental protocols (permit no. 06-08-03).

Data accessibility. Data are available from the Dryad Digital Repository at https://doi.org/10.5061/dryad.vq83bk3pj [40].

Authors' contributions. J.C.S. and G.J.T. designed the shuttlebox; C.D.D., J.C.S. and G.J.T. designed experiments; C.D.D. and J.C.S. performed experiments; D.A.S. analysed results and prepared figures; D.A.S., C.D.D. and G.J.T. wrote the manuscript; D.A.S. and G.J.T. approved the manuscript in its final form.

Competing interests. We declare we have no competing interests.

Funding. This research was funded by Natural Sciences and Engineering Research Council of Canada (NSERC) Discovery grant no. RGPIN-2014-05814 to G.J.T.

Acknowledgements. We are grateful to Viviana Cadena, Jacob Berman, Qian Long and Miriam Richards for assistance with experiments and behavioural assessment, and the Brock Library Open Access Publishing Fund for defraying the costs of publishing.

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
