## [Reviewer comments · Royal Society Open Science]

Review History

RSOS-200416.R0 (Original submission)

Review form: Reviewer 1

Is the manuscript scientifically sound in its present form?

Yes

Are the interpretations and conclusions justified by the results?

No

Is the language acceptable?

Yes

Do you have any ethical concerns with this paper?

No

Have you any concerns about statistical analyses in this paper?

No

Recommendation?

Major revision is needed (please make suggestions in comments)

Comments to the Author(s)

This is an interesting and novel study on the effects of H₂S exposure on thermal preference in zebrafish. The data provides convincing evidence that H₂S exposure causes a downward shift in thermal setpoint in zebrafish. However, the overarching question that the study set out to answer is unclear. It is unclear whether the study is answering the question (1) Are thermal set points manipulated by environmental conditions that limit O₂ supply? or (2) What behavioural strategies do fish use to avoid H₂S toxicity? The introduction discusses aspects of H₂S toxicity which leads me to think that (2) is the question, but reading the opening line of the Discussion (i.e. "A central question in...") leads me to think that (1) is the question. Depending on which question is being asked, the hypothesis could be re-written. Additionally, I do not find the data on the influence of temperature on aquatic surface respiration rates in H₂S convincing. Finally, I do not think the data provides enough direct, compelling evidence to support the conclusion that environmental hijacking of an endogenous gasotransmitter is the mechanism through which H₂S drives the changes in thermal preference seen in the study.

Abstract:

L19: Should read "as measured by reduced rates of aquatic surface respiration." rather than "as measured by reduced rates of aquatic surface respiration rate."

Introduction:

L28-32: The sentence "For instance, application..." does not follow logically from the previous sentence "However, endogenous H₂S...". This should be rewritten for clarity.

L29-30: Many of the roles of endogenous H₂S listed are not relevant to the study.

Providing more background on the ones that are focused on in the discussion (i.e. O₂ sensing) would be more appropriate.

L34: Should read "within their thermal neutral zone" instead of "with their thermal neutral zone"

L47: should read "normoxic conditions" instead of "normoxia conditions"

L49-50: The effects of hypoxia vs H₂S should be elaborated on more here. What do you mean by "exposure to H₂S shares many physiological similarities with hypoxia"?

L53: The hypothesis reads like a prediction.

Methods:

L59-60: It is unclear why fish were habituated to walls lined with white contact paper prior to experiments.

L67-68: How was 0.02% H₂S verified?

L76: How the was 1 hour habitation time determined to be sufficient?

L79: A fairly broad temperature range was used in the shuttlebox experiment.

Could the water temperature have influenced the solubility of H₂S gas?

L81-81: How many fish were removed from the experiment due to distress?

L89-90: It is unclear why you chose 21°C and 28°C as the test temperatures in the methods section. This is not made clear until the end of the discussion. It is also unclear why 2 additional individuals added to the 0% H₂S at 28°C group.

Results:

L117-19: In the results, you state that the probability of ASR increased with H₂S level and with higher temperatures. However, in Fig. 1a) it appears as though the means between the two temperatures within each H₂S treatment are approximately equal, with only an effect of H₂S exposure on ASR rate. Were the means for each temperature within each H₂S treatment significantly different from one another?

L131: This line states that swim velocity increased after gas introduction, including in the testing phase. However, in Fig. 1c iii, swim velocity during the test phase does not appear to increase substantially relative to the ramping phase.

L138-139: Here you state that R=0.54 is reasonably high repeatability. Can you provide a more concrete justification as to why this is considered reasonably high repeatability?

Discussion:

L163-165: Couldn't the blunted response to hypoxia vs. H₂S be due to differences in the degree of metabolic impairment caused by H₂S vs. hypoxia? Elevated H₂S may have caused a more severe reduction in O₂ transport and utilization than the hypoxia level used

L168: It is unclear to me why hypoxia may be necessary for the anapyrexia effects of H₂S in terrestrial animals but not in fish.

L184-185: It seems as though seeking colder temperatures could actually be maladaptive as it could slow the rate of enzymatic H₂S detoxification in the body.

L188-190: I do not think the data provides enough convincing, direct evidence that environmental hijacking of an endogenous gas transmitter is the mechanism through which H₂S drives the changes in thermal preference seen in this study

Review form: Reviewer 2

Is the manuscript scientifically sound in its present form?

Yes

Are the interpretations and conclusions justified by the results?

Yes

Is the language acceptable?

Yes

Do you have any ethical concerns with this paper?

No

Have you any concerns about statistical analyses in this paper?

No

Recommendation?

Major revision is needed (please make suggestions in comments)

Comments to the Author(s)

I have reviewed the article "Hydrogen sulfide exposure reduces thermal set point in zebrafish". Hydrogen sulfide is an inhibitor of aerobic respiration. The authors use an ectotherm to address whether exposure to exogenous H₂S induces a reduction in the internal temperature set point, hypothesizing that H₂S exposure would lower the temperature preference of the fish over the course of the experiment. The authors find a clear effect of H₂S which reduced temperature preference and swimming velocity.

I think the writing for the manuscript is clear and succinct, and the results for this study are clearly presented. My main questions relate to the experimental set-up that will require clarification in the methods section. I also have some minor editing comments listed.

Main comments:

- Line 67- 72: Bubbling mixed gases still doesn't guarantee that the hydrogen sulfide stays in the water. Can the authors clarify whether the final water concentrations of H₂S and O₂ levels were monitored in the exposure tank during the experiment at all? Was there at least a pilot test to determine if the desired H₂S concentration would be maintained using this method?
- Line 72 and 119-121: Regarding the Plexiglas cover over the tank, was that right on the surface of the water inside the tank? Or was there some space between that gas is being pumped into? I can see how that would help to maintain gas pressures and to prevent condensation, but I don't see how that would avoid gas gradients from forming in the tank water. Were there pumps to help circulate and mix the water at all?
- Line 81-82: It's mentioned that fish that were in distress were removed. Were these fish put into a recovery tank and were they recovered fully? How many lost equilibrium over the 4hr period?

- Was O₂ of the water measured during the period of experiment? Could the surface layer have been more aerated?

Minor comments:

- "Set point" is two words in some places and one word in others.
- Figure 1B (i) and (ii) are not labelled
- What is the top figure in (ii) in B?
- Line 30: These two sentence appear to be disconnected. The previous sentence talks about the physiological role of endogenous H₂S, and this sentence lists an example of the effects of exogenous H₂S.
- Line 34: "... performed in small mammals *within* their thermoneutral zone,..."
- Line 51: It's worth mentioning that H₂S inhibits the O₂ binding enzyme cytochrome oxidase in mitochondria, which also explains why the response to H₂S is similar to hypoxia.
- Line 127: "These observations point to *the* defense ..."
- Line 168: O₂ and H₂S concentrations have an inverse relationship (i.e. higher the H₂S, lower the O₂)
- Line 182: "... will depend on the *level* of environmental H₂S..."
- Line 182: I wouldn't view this as maladaptive for the fish (depending on how H₂S and hypoxia-tolerant they are) if it's just that cold temperatures can stimulate more H₂S to be released into the mud and change their environment.
- Line 187-188: Whether it was due to the role of H₂S in O₂ sensing wasn't something that this study addressed.
- Bibliography: check consistency in formatting of the paper titles

Decision letter (RSOS-200416.R0)

Dear Dr Tattersall,

The editors assigned to your paper ("Hydrogen sulfide exposure reduces thermal set point in zebrafish") have now received comments from reviewers. We would like you to revise your paper in accordance with the referee and Associate Editor suggestions which can be found below (not including confidential reports to the Editor). Please note this decision does not guarantee eventual acceptance.

Please submit a copy of your revised paper before 10-May-2020. Please note that the revision deadline will expire at 00.00am on this date. If we do not hear from you within this time then it will be assumed that the paper has been withdrawn. In exceptional circumstances, extensions may be possible if agreed with the Editorial Office in advance. We do not allow multiple rounds of revision so we urge you to make every effort to fully address all of the comments at this stage. If deemed necessary by the Editors, your manuscript will be sent back to one or more of the original reviewers for assessment. If the original reviewers are not available, we may invite new reviewers.

- Data accessibility

If you wish to submit your supporting data or code to Dryad (<http://datadryad.org/>), or modify your current submission to dryad, please use the following link:
<http://datadryad.org/submit?journalID=RSOS&manu=RSOS-200416>

- Competing interests

- Authors' contributions

- Acknowledgements

- Funding statement

Best regards,
Lianne Parkhouse
Editorial Coordinator
Royal Society Open Science.
openscience@royalsociety.org

on behalf of Dr Michael Tobler (Associate Editor) and Pete Smith (Subject Editor)
openscience@royalsociety.org

Associate Editor's comments (Dr Michael Tobler):

We have received the feedback from two reviewers that agree that the study is interesting and makes a potentially novel contribution to the literature. However, they also highlight a number of concerns regarding the experimental design and the interpretation of data that will need to be addressed before the manuscript can be considered for publication.

Reviewers' Comments to Author:

Reviewer: 1
Comments to the Author(s)

This is an interesting and novel study on the effects of H₂S exposure on thermal preference in zebrafish. The data provides convincing evidence that H₂S exposure causes a downward shift in thermal setpoint in zebrafish. However, the overarching question that the study set out to answer is unclear. It is unclear whether the study is answering the question (1) Are thermal set points manipulated by environmental conditions that limit O₂ supply? or (2) What behavioural strategies do fish use to avoid H₂S toxicity? The introduction discusses aspects of H₂S toxicity which leads me to think that (2) is the question, but reading the opening line of the Discussion (i.e. "A central question in...") leads me to think that (1) is the question. Depending on which question is being asked, the hypothesis could be re-written. Additionally, I do not find the data on the influence of temperature on aquatic surface respiration rates in H₂S convincing. Finally, I do not think the data provides enough direct, compelling evidence to support the conclusion that environmental hijacking of an endogenous gasotransmitter is the mechanism through which H₂S drives the changes in thermal preference seen in the study.

Abstract:

L19: Should read "as measured by reduced rates of aquatic surface respiration." rather than "as measured by reduced rates of aquatic surface respiration rate."

Introduction:

L28-32: The sentence "For instance, application..." does not follow logically from the previous sentence "However, endogenous H₂S...". This should be rewritten for clarity.

L29-30: Many of the roles of endogenous H₂S listed are not relevant to the study.

Providing more background on the ones that are focused on in the discussion (i.e. O₂ sensing) would be more appropriate.

L34: Should read "within their thermal neutral zone" instead of "with their thermal neutral zone"

L47: should read "normoxic conditions" instead of "normoxia conditions"

L49-50: The effects of hypoxia vs H₂S should be elaborated on more here. What do you mean by “exposure to H₂S shares many physiological similarities with hypoxia”?

L53: The hypothesis reads like a prediction.

Methods:

L59-60: It is unclear why fish were habituated to walls lined with white contact paper prior to experiments.

L67-68: How was 0.02% H₂S verified?

L76: How was the 1 hour habitation time determined to be sufficient?

L79: A fairly broad temperature range was used in the shuttlebox experiment.

Could the water temperature have influenced the solubility of H₂S gas?

L81-81: How many fish were removed from the experiment due to distress?

L89-90: It is unclear why you chose 21°C and 28°C as the test temperatures in the methods section. This is not made clear until the end of the discussion. It is also unclear why 2 additional individuals were added to the 0% H₂S at 28°C group.

Results:

L117-119: In the results, you state that the probability of ASR increased with H₂S level and with higher temperatures. However, in Fig. 1a) it appears as though the means between the two temperatures within each H₂S treatment are approximately equal, with only an effect of H₂S exposure on ASR rate. Were the means for each temperature within each H₂S treatment significantly different from one another?

L131: This line states that swim velocity increased after gas introduction, including in the testing phase. However, in Fig. 1c iii, swim velocity during the test phase does not appear to increase substantially relative to the ramping phase.

L138-139: Here you state that R=0.54 is reasonably high repeatability. Can you provide a more concrete justification as to why this is considered reasonably high repeatability?

Discussion:

L163-165: Couldn't the blunted response to hypoxia vs. H₂S be due to differences in the degree of metabolic impairment caused by H₂S vs. hypoxia? Elevated H₂S may have caused a more severe reduction in O₂ transport and utilization than the hypoxia level used.

L168: It is unclear to me why hypoxia may be necessary for the anapyrexia effects of H₂S in terrestrial animals but not in fish.

L184-185: It seems as though seeking colder temperatures could actually be maladaptive as it could slow the rate of enzymatic H₂S detoxification in the body.

L188-190: I do not think the data provides enough convincing, direct evidence that environmental hijacking of an endogenous neurotransmitter is the mechanism through which H₂S drives the changes in thermal preference seen in this study.

Reviewer: 2

Comments to the Author(s)

I have reviewed the article “Hydrogen sulfide exposure reduces thermal set point in zebrafish”. Hydrogen sulfide is an inhibitor of aerobic respiration. The authors use an ectotherm to address whether exposure to exogenous H₂S induces a reduction in the internal temperature set point, hypothesizing that H₂S exposure would lower the temperature preference of the fish over the course of the experiment. The authors find a clear effect of H₂S which reduced temperature preference and swimming velocity.

I think the writing for the manuscript is clear and succinct, and the results for this study are clearly presented. My main questions relate to the experimental set-up that will require clarification in the methods section. I also have some minor editing comments listed.

Main comments:

- Line 67- 72: Bubbling mixed gases still doesn't guarantee that the hydrogen sulfide stays in the water. Can the authors clarify whether the final water concentrations of H₂S and O₂ levels were monitored in the exposure tank during the experiment at all? Was there at least a pilot test to determine if the desired H₂S concentration would be maintained using this method?

- Line 72 and 119-121: Regarding the Plexiglas cover over the tank, was that right on the surface of the water inside the tank? Or was there some space between that gas is being pumped into? I can see how that would help to maintain gas pressures and to prevent condensation, but I don't see how that would avoid gas gradients from forming in the tank water. Were there pumps to help circulate and mix the water at all?
- Line 81-82: It's mentioned that fish that were in distress were removed. Were these fish put into a recovery tank and were they recovered fully? How many lost equilibrium over the 4hr period?
- Was O₂ of the water measured during the period of experiment? Could the surface layer have been more aerated?

Minor comments:

- "Set point" is two words in some places and one word in others.
- Figure 1B (i) and (ii) are not labelled
- What is the top figure in (ii) in B?
- Line 30: These two sentence appear to be disconnected. The previous sentence talks about the physiological role of endogenous H₂S, and this sentence lists an example of the effects of exogenous H₂S.
- Line 34: "... performed in small mammals *within* their thermoneutral zone,..."
- Line 51: It's worth mentioning that H₂S inhibits the O₂ binding enzyme cytochrome oxidase in mitochondria, which also explains why the response to H₂S is similar to hypoxia.
- Line 127: "These observations point to *the* defense ..."
- Line 168: O₂ and H₂S concentrations have an inverse relationship (i.e. higher the H₂S, lower the O₂)
- Line 182: "... will depend on the *level* of environmental H₂S..."
- Line 182: I wouldn't view this as maladaptive for the fish (depending on how H₂S and hypoxia-tolerant they are) if it's just that cold temperatures can stimulate more H₂S to be released into the mud and change their environment.
- Line 187-188: Whether it was due to the role of H₂S in O₂ sensing wasn't something that this study addressed.
- Bibliography: check consistency in formatting of the paper titles

Author's Response to Decision Letter for (RSOS-200416.R0)

See Appendix A.

RSOS-200416.R1 (Revision)

Review form: Reviewer 1

Is the manuscript scientifically sound in its present form?

Yes

Are the interpretations and conclusions justified by the results?

Yes

Is the language acceptable?

Yes

Do you have any ethical concerns with this paper?

No

Have you any concerns about statistical analyses in this paper?

No

Recommendation?

Accept with minor revision (please list in comments)

Comments to the Author(s)

The authors have addressed most of my comments in both the rebuttal as well as in the manuscript. In the process, they have greatly increased the quality of their manuscript. I only have a few additional comments.

Methods:

In a response to Reviewer 2, you mention that pilot studies were performed that indicated that H₂S levels were constant throughout the aeration period. These experiments should be noted in the methods alongside the H₂S levels measured in these experiments.

Water pH can dramatically influence the proportion of H₂S that is present in its toxic form. What was the water pH in the shuttlebox chambers during the behavioural trials?

Results:

Line 171: should read "consistent" rather than "consist"

Discussion:

In the rebuttal, you mention that the solubility of H₂S changes quite dramatically with temperature, which may, in turn, influence the motivation of fish to seek colder environments and/or perform ASR. This caveat should be appropriately identified and addressed in the discussion.

Line 183-185: It is unclear to me how the change in body temperature is consistent with the involvement of H₂S in O₂ sensing by NECs.

Line 187-188: In the rebuttal, you give a good explanation as to why it is unclear whether hypoxia is required for the anapyrexia effects of H₂S in mammals. However, because no changes were made to the manuscript, it is still unclear to the reader why hypoxia may be necessary for the anapyrexia effects of H₂S. Providing additional details in the text would help to alleviate this issue as well as help to highlight the novelty of your data.

Minor:

Throughout the text, "O₂" and "oxygen" are used interchangeably. One should be chosen and used consistently throughout the text.

Decision letter (RSOS-200416.R1)

Dear Dr Tattersall

On behalf of the Editors, we are pleased to inform you that your Manuscript RSOS-200416.R1 "Hydrogen sulfide exposure reduces thermal set point in zebrafish" has been accepted for publication in Royal Society Open Science subject to minor revision in accordance with the referees' reports. Please find the referees' comments along with any feedback from the Editors below my signature.

We invite you to respond to the comments and revise your manuscript. Below the referees' and Editors' comments (where applicable) we provide additional requirements. Final acceptance of

your manuscript is dependent on these requirements being met. We provide guidance below to help you prepare your revision.

Please submit your revised manuscript and required files (see below) no later than 7 days from today's (ie 10-Sep-2020) date. Note: the ScholarOne system will 'lock' if submission of the revision is attempted 7 or more days after the deadline. If you do not think you will be able to meet this deadline please contact the editorial office immediately.

Best regards,

on behalf of the Associate Editor, and Professor Pete Smith (Subject Editor)
openscience@royalsociety.org

Reviewer comments to Author:

Reviewer: 1
Comments to the Author(s)

The authors have addressed most of my comments in both the rebuttal as well as in the manuscript. In the process, they have greatly increased the quality of their manuscript. I only have a few additional comments.

Methods:

In a response to Reviewer 2, you mention that pilot studies were performed that indicated that H₂S levels were constant throughout the aeration period. These experiments should be noted in the methods alongside the H₂S levels measured in these experiments.

Water pH can dramatically influence the proportion of H₂S that is present in its toxic form. What was the water pH in the shuttlebox chambers during the behavioural trials?

Results:

Line 171: should read "consistent" rather than "consist"

Discussion:

In the rebuttal, you mention that the solubility of H₂S changes quite dramatically with temperature, which may, in turn, influence the motivation of fish to seek colder environments and/or perform ASR. This caveat should be appropriately identified and addressed in the discussion.

Line 183-185: It is unclear to me how the change in body temperature is consistent with the involvement of H₂S in O₂ sensing by NECs.

Line 187-188: In the rebuttal, you give a good explanation as to why it is unclear whether hypoxia is required for the anapyrexia effects of H₂S in mammals. However, because no changes were

made to the manuscript, it is still unclear to the reader why hypoxia may be necessary for the anapyrexia effects of H₂S. Providing additional details in the text would help to alleviate this issue as well as help to highlight the novelty of your data.

Minor:

Throughout the text, “O₂” and “oxygen” are used interchangeably. One should be chosen and used consistently throughout the text.

===PREPARING YOUR MANUSCRIPT===

- one version identifying all the changes that have been made (for instance, in coloured highlight, in bold text, or tracked changes);
- a 'clean' version of the new manuscript that incorporates the changes made, but does not highlight them. This version will be used for typesetting.

===PREPARING YOUR REVISION IN SCHOLARONE===

- 1) One version identifying all the changes that have been made (for instance, in coloured highlight, in bold text, or tracked changes);
 - 2) A 'clean' version of the new manuscript that incorporates the changes made, but does not highlight them.
 - An individual file of each figure (EPS or print-quality PDF preferred [either format should be produced directly from original creation package], or original software format).
 - An editable file of each table (.doc, .docx, .xls, .xlsx, or .csv).
 - An editable file of all figure and table captions.
- Note: you may upload the figure, table, and caption files in a single Zip folder.
- Any electronic supplementary material (ESM).
 - If you are requesting a discretionary waiver for the article processing charge, the waiver form must be included at this step.
 - If you are providing image files for potential cover images, please upload these at this step, and inform the editorial office you have done so. You must hold the copyright to any image provided.
 - A copy of your point-by-point response to referees and Editors. This will expedite the preparation of your proof.

- Ensure that your data access statement meets the requirements at <https://royalsociety.org/journals/authors/author-guidelines/#data>. You should ensure that you cite the dataset in your reference list. If you have deposited data etc in the Dryad repository, please only include the 'For publication' link at this stage. You should remove the 'For review' link.
- If you are requesting an article processing charge waiver, you must select the relevant waiver option (if requesting a discretionary waiver, the form should have been uploaded at Step 3 'File upload' above).
- If you have uploaded ESM files, please ensure you follow the guidance at <https://royalsociety.org/journals/authors/author-guidelines/#supplementary-material> to include a suitable title and informative caption. An example of appropriate titling and captioning may be found at [https://figshare.com/articles/Table_S2_from_Is_there_a_trade-off_between_peak_performance_and_performance_breadth_across_temperatures_for_aerobic_sc ope_in_teleost_fishes_/3843624](https://figshare.com/articles/Table_S2_from_Is_there_a_trade-off_between_peak_performance_and_performance_breadth_across_temperatures_for_aerobic_scope_in_teleost_fishes_/3843624).

Author's Response to Decision Letter for (RSOS-200416.R1)

See Appendix B.

Decision letter (RSOS-200416.R2)

Dear Dr Tattersall,

It is a pleasure to accept your manuscript entitled "Hydrogen sulfide exposure reduces thermal set point in zebrafish" in its current form for publication in Royal Society Open Science. Please ensure that you send to the editorial office an editable version of your accepted manuscript, and individual files for each figure and table included in your manuscript. You can send these in a zip folder if more convenient. Failure to provide these files may delay the processing of your proof. You may disregard this request if you have already provided these files to the editorial office.

on behalf Pete Smith (Subject Editor)
openscience@royalsociety.org

Appendix A

Reviewer comments

Author Responses in Bold

Reviewer: 1

This is an interesting and novel study on the effects of H₂S exposure on thermal preference in zebrafish. The data provides convincing evidence that H₂S exposure causes a downward shift in thermal setpoint in zebrafish. However, the overarching question that the study set out to answer is unclear. It is unclear whether the study is answering the question (1) Are thermal set points manipulated by environmental conditions that limit O₂ supply? or (2) What behavioural strategies do fish use to avoid H₂S toxicity? The introduction discusses aspects of H₂S toxicity which leads me to think that (2) is the question, but reading the opening line of the Discussion (i.e. "A central question in...") leads me to think that (1) is the question. Depending on which question is being asked, the hypothesis could be re-written. Additionally, I do not find the data on the influence of temperature on aquatic surface respiration rates in H₂S convincing. Finally, I do not think the data provides enough direct, compelling evidence to support the conclusion that environmental hijacking of an endogenous gasotransmitter is the mechanism through which H₂S drives the changes in thermal preference seen in the study.

Thank you for the thoughtful summary and comments. Regarding the point about possible conflicting objectives, we did not intend these to be disparate. Indeed, the objectives (last paragraph of the introduction) are focused on the influence of H₂S on thermal set points, not H₂S toxicity, so our returning to point 1 (above) in the discussion is consistent with a writing style of re-addressing context and objectives up front in the discussion. Mentioning H₂S as a toxin that animals may have behavioural responses to is not a topic we can shy away from, however, and thus the outline of our introduction flowed from general, concept based principles (H₂S, environmental parameters, etc) and placing the paper into a wider context, down through neurophysiological concepts, and toward the main objective regarding thermal set-points (i.e. behavioural strategies fish might employ).

So, we don't really see these as mutually exclusive ideas. Indeed, choosing a lower set-point (q1 above) has to be achieved behaviourally in an ectotherm, and thus set-points and behaviour are integrated concepts in ectotherm thermoregulatory responses. Although the beginning of the discussion starts with "a central question in thermoregulatory physiology", this is not "our central question was...", so perhaps this wording gave an incorrect impression? The concept of thermal set-points itself has supporters and detractors, which is why we mention it as a 'central question' (i.e. an existential question perhaps?), since it is difficult to resolve in studies of endotherms, but possible to resolve in ectotherms that employ behaviour to manipulate body temperature.

We have reviewed our presentation of the Aquatic Surface Respiration behaviour with respect to temperature, and agree that the change in ASR probability was not well explained. In addition to our overall reorganisation of the manuscript flow, we have included in the new Figure 2, a plot of the effect sizes. The temperature effect indicates an ~3.5-fold increase in ASR with temperature (0.11 to 2.43, excludes log OR =0). This change does not equal the impact of H₂S, indicating that temperature does offset the effects of 0.02% H₂S. However, we note that this is 3.5-fold for only the average 7°C average temperature change. Our results lead to the prediction of smaller changes in the individuals that exhibited the least change, and higher ASR probability in the individuals exhibiting the largest temperature changes. In the comments below, and revised manuscript these effects have hopefully been more clearly explained.

As for the conclusion that hijacking of endogenous gasotransmission is occurring, we have tempered this from a conclusion to a proposed mechanism (please see final sentence of the revised discussion). Indeed, we were planning experiments to follow-up on this in the future.

Abstract:

L19: Should read “as measured by reduced rates of aquatic surface respiration.” rather than “as measured by reduced rates of aquatic surface respiration rate.”

Thank you. We have changed this sentence (by deleting the redundant “rate”)

Introduction:

L28-32: The sentence “For instance, application...” does not follow logically from the previous sentence “However, endogenous H₂S...”. This should be rewritten for clarity.

Thank you. We have removed the “for instance”, which allows the sentence to connect more clearly with the previous sentences. Essentially, we are discussing the various roles that H₂S has been shown to exhibit, so the ‘for instance’ was not needed.

L29-30: Many of the roles of endogenous H₂S listed are not relevant to the study. Providing more background on the ones that are focused on in the discussion (i.e. O₂ sensing) would be more appropriate.

The roles we mentioned highlighted that H₂S is not simply a toxin, but has sublethal and long-term impacts. Synaptic activity and cognitive function are important for problem solving tasks and behaviours (paramount for operant conditioning), and inflammation is strongly linked to thermoregulation (just no space to detail here). To further inform the reader of the metabolic roles, we have added references to the oxygen sensing role with the following change to the introduction:

However, H₂S is not exclusively toxic and has endogenous roles including the physiological response to hypoxia and regulation of synaptic activity, cognitive function, inflammation, and oxygen sensing [4–6].”

L34: Should read “within their thermal neutral zone” instead of “with their thermal neutral zone”

Thank you. We have changed “with” to “within” which fixes that sentence.

L47: should read “normoxic conditions” instead of “normoxia conditions”

Thank you. We have changed “normoxia” to “normoxic” as suggested.

L49-50: The effects of hypoxia vs H₂S should be elaborated on more here. What do you mean by “exposure to H₂S shares many physiological similarities with hypoxia”?

L53: The hypothesis reads like a prediction.

We have reworded this to:

“We tested the hypothesis that H₂S triggers a reduction in individual thermal set point, pointing to sublethal effects of H₂S on physiology and behaviour.”

Methods:

L59-60: It is unclear why fish were habituated to walls lined with white contact paper prior to experiments.

Since we used camera tracking that required a constant contrast with background, the test chamber had to be lined with a light background to capture the darker pigmented fish and to minimise shadow-based artifacts. This required that we would transfer fish from a holding tank to

a novel environment with additional complications associated with the stress response, since many fish are known to 'blanch' (i.e. melanosomes in melanocytes contract) as a background colour matching response and this colour change is mediated through catecholamines. We reasoned that the stress of novelty would affect behaviour adversely, so it was the responsible thing to change the holding tank conditions to remove this potential stressor. There is evidence in teleosts that housing in tanks with black background leads to greater cortisol release when undergoing noise stress and handling procedures as well:

https://joe.bioscientifica.com/view/journals/joe/105/1/joe_105_1_013.xml).

We have tried to make it clear that housing conditions were designed to match experimental conditions, to avoid additional confounding effects.

New wording:

“Fish were housed at least 24 days and habituated to walls lined with white contact paper (required for the automatic camera tracking software) prior to experiments, to mitigate the stress of a change in visual environment. Moreover, dark walls may facilitate stress responses that affect subsequent behavioural trials [27].”

L67-68: How was 0.02% H₂S verified?

H₂S was mixed volumetrically using two calibrated, rotameter flowmeters (i.e. flow rates calibrated against a certified volumeter we have in the lab). We had a certified 0.2% H₂S tank from Praxair and used rotameter volumeter flowmeters to mix compressed air with 0.2% H₂S and diluted the gas this way. This is a standard means to mix calibration gases and is highly robust, since the mixing procedure does not rely on electronic sensors that drift over time. Indeed, this approach to gas mixing is used to generate calibration gases. Since we used paired rotameter flowmeters to first mix the gas prior to splitting and then bubbled the mixed gas into the respective side chambers of the shuttle box, the total H₂S levels were robust, although we have expressed them throughout as percentages to reflect the fact that gases were mixed using this approach. But measuring H₂S itself is highly challenging due to the 3 different states the dissolved substance can exist as a function of pH. So, our volumetric approach was adopted in order to achieve consistency.

New wording:

“Air and 0.2% H₂S (Praxair certified) were mixed volumetrically to achieve the appropriate H₂S concentration (0% or 0.02% H₂S) using two calibrated flow meters (Omega rotameters) to achieve a total constant flow of 5000 mL min⁻¹ (0.07% H₂S elicited severe distress, not shown).”

L76: How the was 1 hour habitation time determined to be sufficient?

Behavioural responses to the chamber alone were validated in prior experiments at fixed temperatures. This control allowed us to determine the length of time necessary for the fish to habituate to a constant novel environment. Smooth functions of the shuttling rate show that fish habituated within about one hour (new Supplementary Figure 2) in both constant conditions and in the experiment proper. The capacity of both the chamber and of zebrafish to exhibit the expected thermoregulatory behaviours in response to a known inducer of anapnoea, hypoxia, were also validated in prior experiments. These controls have been detailed in the Methods section.

L79: A fairly broad temperature range was used in the shuttlebox experiment. Could the water temperature have influenced the solubility of H₂S gas?

The nature of the experiment is that the fish determine their own temperature, and each chamber is being continuously flooded with H₂S gas in the aqueous phase, so to characterise that a broad

temperature range was used rightly places the responsibility on the fish. Constraining to a narrow, prescribed range would have precluded us from testing the hypothesis of a change in set point since we might have truncated the range of possible temperature preferences. The fish select those temperatures based on learned and innate behaviours and drive the experiment. Gases diffuse according to their partial pressures (Fick's Law of Diffusion), rather than their concentration, so diffusion across the gills would be responding according to similar partial pressure gradients at any temperature.

At normal atmospheric pressures (like we used), H₂S gas acts according to Henry's laws (<https://webbook.nist.gov/cgi/cbook.cgi?ID=C7783064&Mask=10#Notes>) and appears to have normal fugacity (the factor you need to multiply Henry's k coefficients by to get the "real activity" of a gas – in other words, normal fugacity means the gas is easy enough to get into solution, and isn't escaping from the water without first equilibrating within the water).

See also: https://www.gasliquids.com/pdfs/1989_SolubilityHydrogenSulphideInWater.pdf

Henry's k for 15 to 35°C changes from a value of 42 to 60, so the solubility changes by 42% across the extremes of the experiment (rarely achieved by the fish). Just like other gases there will be differences in solubility with temperature, but this is something one cannot control, and the difference in solubility across a 7°C temperature difference (i.e. 28°C vs 21°C thermal preference change between 0% H₂S and 0.02% H₂S) is far smaller than the difference between 0% H₂S water and 0.02% H₂S water, so how to incorporate solubility differences into the scope of the study is not clear. Fundamentally, the point the reviewer makes is valid, but from the perspective of the fish behaviour, the trade-off between cooling down vs. exposure to elevated concentrations might not be so severe a trade-off, since the diffusion of gas into the body would be driven by a constant partial pressure gradient.

L81-81: How many fish were removed from the experiment due to distress?

In our attempt at the brief wording requirements during the initial submission to Biology Letters, the wording here did not accurately depict the study's inclusionary criteria. In accord with our animal care protocol, on a precautionary principle, we terminated any experiment where fish showed any indication of unusual swimming behaviour. Only 2 fish lost equilibrium in either condition. In revisiting this data, we discovered in the experiment notes that several experiments were terminated early due simply to an excess amount of time at the surface (performing ASR?), which might have interfered with their motivation to perform shuttling in this experiment. Their termination times are marked in Figure 1. For these individuals, we have taken the average of their testing-phase responses up to the termination time. To determine whether premature removal may have biased the results (e.g., a survivor bias), we reformed all analyses with a testing period of 2 hours matched to the ramping period. This shorter time frame reduces disparities in the period over which averaged responses are collected. We do not find our results qualitatively impacted by this reduced period. The reanalysed results are provided in the revised supplementary statistical tables.

L89-90: It is unclear why you chose 21°C and 28°C as the test temperatures in the methods section. This is not made clear until the end of the discussion. It is also unclear why 2 additional individuals added to the 0% H₂S at 28°C group.

The 28 and 21°C choice was chosen based on the mean thermal preference. In retrospect, we misjudged the order of presentation of the two experiments, which should have been in study chronological order. We have placed ASR in its own Figure 2, along with the modeled effects of the predictors. We expect this will improve the clarity of the results and allow readers to understand the logical flow of ideas, since the ASR experiments followed after the thermoregulation experiments. The experiments were designed to formally test a) our anecdotal observations of increased ASR during H₂S experiments in the shuttle box, and b) whether the

mean selected temperatures influenced the ASR. Because the final data had not been fully analysed when the ASR experiments were initiated, 28 and 21°C were chosen by rounding to nearest whole value temperature readings.

Two additional individuals are present in the 0% H₂S group because using all available data helps with estimating model variances, even when unpaired. These fish simply did not get tested at the 0.02% H₂S, likely due to student time constraints.

Results:

L117-19: In the results, you state that the probability of ASR increased with H₂S level and with higher temperatures. However, in Fig. 1a) it appears as though the means between the two temperatures within each H₂S treatment are approximately equal, with only an effect of H₂S exposure on ASR rate. Were the means for each temperature within each H₂S treatment significantly different from one another?

Examining plots like these also requires including the paired nature of the effect, which are usually difficult to assess graphically. However, we find that our presentation of the results were not well-developed, and we understand why they have caused confusion. We cannot use means for these data, partly because the data are fractional (percentage) data, but also because mean would not reflect the central tendency.

We have revised the ASR analysis and description in a new Figure 2 and in the text to depict the changes to explicitly show the log odds ratio, estimates and revised the text to discuss the effect of temperature in terms of odds. We report a ~3.5-fold increase in ASR over the 7°C change in temperature. We judge this to be not only statistically significant (due to the 95% credible intervals not including zero) but also biologically significant. We also have added discussion that this change in ASR will differentially impact individuals that select relatively much colder or warmer temperatures than the means. Further investigation of this phenomenon is planned in the senior author's laboratory.

L131: This line states that swim velocity increased after gas introduction, including in the testing phase. However, in Fig. 1c iii, swim velocity during the test phase does not appear to increase substantially relative to the ramping phase.

The purpose of analysing swim velocity was to determine whether the H₂S causes behavioural deficits that would impair our ability to determine whether H₂S influences the thermal setpoint. A greatly reduced velocity might point to impaired fish locomotion that causes the fish to spend more time on the cold side because of their poikilothermic dependence on temperature. Rather, we find statistical evidence that velocity is increased (which is noted in the results), but even a constant velocity would reject the alternative hypothesis of decreased performance.

We have added a note in the text that,

“Thus, we do not find evidence for behavioural impairment that may have caused this side preference.”

L138-139: Here you state that R=0.54 is reasonably high repeatability. Can you provide a more concrete justification as to why this is considered reasonably high repeatability?

Working from the analogy that Repeatability (or Intra Class Correlation Coefficients) are analogous to Pearson's r values, according to Cohen (1988), the effect size is low if the value of r varies around 0.1, medium if r varies around 0.3, and large if r varies more than 0.5.

Please see:

https://www.researchgate.net/publication/305699288_Effect_size_guidelines_for_individual_differences_researchers

but then for interpreting R see: https://en.wikipedia.org/wiki/Intraclass_correlation

Cicchetti (1994) gives the following often quoted guidelines for interpretation for ICC inter-rater agreement measures:

- Less than 0.40—poor.
- Between 0.40 and 0.59—fair.
- Between 0.60 and 0.74—good.
- Between 0.75 and 1.00—excellent.

A different guideline is given by Koo and Li (2016)

- below 0.50: poor
- between 0.50 and 0.75: moderate
- between 0.75 and 0.90: good
- above 0.90: excellent

So, all considered we have decided to re-word this to “moderate repeatability”. Thank you for bringing this to our attention.

Discussion:

L163-165: Couldn't the blunted response to hypoxia vs. H₂S be due to differences in the degree of metabolic impairment caused by H₂S vs. hypoxia? Elevated H₂S may have caused a more severe reduction in O₂ transport and utilization than the hypoxia level used

We agree with the reviewer that there is a lack of evidence for this comparison and have removed it. Our comparison was overly simplistic, anyway. Even if both responses acted via similar pathways, there are bound to be different dose-dependencies. Non-linear/threshold responses in hypoxia are common, but it is not clear how H₂S would work. If operating as a gasotransmitter, it is likely that H₂S effects would be linear with respect to [H₂S].

The general results of our preliminary hypoxia treatments were useful for demonstrating that the shuttlebox can recapitulate known fish behaviours (i.e. hypoxia induced behavioural anapyrexia has been shown in trout and cod – these papers are now cited) before testing novel responses and facilitate examining otherwise established changes in behavioural thermal set-points.

L168: It is unclear to me why hypoxia may be necessary for the anapyretic effects of H₂S in terrestrial animals but not in fish.

We have re-worded this sentence by correctly referring to mammals, not to all terrestrial mammals, since the papers cited were done in mice. We do not necessarily think that hypoxia is necessary in mammals, but the previous studies did not eliminate that possibility (i.e. they only demonstrated anapyrexia when using H₂S in combination with hypoxia and cold exposure). It is unclear why the original authors used hypoxia with H₂S, which tempers the scope of conclusions that could be drawn from those studies. Since hypoxia was already an established anapyretic signal, it simply wasn't entirely clear from the mammalian work if H₂S was activating alone or enhanced by the hypoxic stimulus, a point not raised in those original papers. Thus, the current work was conceived in part as a test of whether hypoxia is more generally required. We expect our current work will drive the development of more refined hypotheses about the role of H₂S in body temperature regulation.

L184-185: It seems as though seeking colder temperatures could actually be maladaptive as it could slow the rate of enzymatic H₂S detoxification in the body.

We have now pointed to relevant literature that a) reducing core temperature reduces toxicity effects and may be why animals have evolved the capacity for anapyrexia, and b) low temperature prolongs the duration of toxicity, but reduces its magnitude (i.e., flattens the curve).

The text now includes,

Thermoregulation can help reduce toxicity by reducing the magnitude of effects despite prolonging duration, and this action may be a central force in the evolution of anapyrexia [38,39].

L188-190: I do not think the data provides enough convincing, direct evidence that environmental hijacking of an endogenous gasotransmitter is the mechanism through which H₂S drives the changes in thermal preference seen in this study

We have tempered the writing to suggest this as a plausible mechanism. We recognise that there may also or instead be direct metabolic effects on tissues or gasotransmitter-driven responses. However, because fish could still swim normally AND make operant conditioning decisions (thought provoking, itself), we think it less likely that there is impairment or that metabolic poisoning explains the results. This is the subject of experiments underway in the senior author's laboratory.

We have changed the wording here from “demonstrating” to “suggesting”.

Reviewer: 2

Comments to the Author(s)

I have reviewed the article "Hydrogen sulfide exposure reduces thermal set point in zebrafish". Hydrogen sulfide is an inhibitor of aerobic respiration. The authors use an ectotherm to address whether exposure to exogenous H₂S induces a reduction in the internal temperature set point, hypothesizing that H₂S exposure would lower the temperature preference of the fish over the course of the experiment. The authors find a clear effect of H₂S which reduced temperature preference and swimming velocity. I think the writing for the manuscript is clear and succinct, and the results for this study are clearly presented. My main questions relate to the experimental set-up that will require clarification in the methods section. I also have some minor editing comments listed.

Main comments:

- Line 67- 72: Bubbling mixed gases still doesn't guarantee that the hydrogen sulfide stays in the water. Can the authors clarify whether the final water concentrations of H₂S and O₂ levels were monitored in the exposure tank during the experiment at all? Was there at least a pilot test to determine if the desired H₂S concentration would be maintained using this method?

Given that our system was a circulating system with constantly, mixed gas aeration, this is in fact the most effective way to ensure that water is appropriately equilibrated with gas, so we are confused a little by this comment, but think that an enhanced description of our setup might help matters (we have included a description in the supplementary materials, also referring to our original paper in 2012 that describes the set-up as well).

Indeed, if one were to calibrate an oxygen electrode, you would use this approach since gases equilibrate in solution when constantly aerated and mixed. Since the aeration was continuous, and diffusion out of water is far slower than convective mixing into water, the expectation that the gas escapes too fast from the water is not appropriate. We did purchase an H₂S electrode system to test water H₂S, but discovered that the calibration solutions only mimic H₂S (i.e., you don't actually use 0% H₂S and 0.02% H₂S solution as standards) because of the difficulties in accurately measuring H₂S in solution electrochemically, so this solution to verifying [H₂S] was unsatisfactory. Pilot studies did indicate that constant H₂S levels were present throughout a period of aeration, but we did not record these values as a matter of routine for the behavioural results since sampling the water would have disturbed the fish. Rather, we used buffered water to ensure that water pH did not change profoundly during behaviour trials, which in combination with accurate mixing should ensure constant levels throughout a trial. Since this was a behaviour paper rather than a chemical analysis study, we elected to use the far more reliable volumetric dilution method. Fortunately, because Praxair certifies their gas mixtures, we can easily dilute the flow rate by 1:10 ratio. Since we have a Volumeter gas calibrating system, we are always able to check our flowmeters against 1st principles.

- Line 72 and 119-121: Regarding the Plexiglas cover over the tank, was that right on the surface of the water inside the tank? Or was there some space between that gas is being pumped into? I can see how that would help to maintain gas pressures and to prevent condensation, but I don't see how that would avoid gas gradients from forming in the tank water. Were there pumps to help circulate and mix the water at all?

Good question. It was not right on the surface. We had small spacers (plasticine) to allow the mixed gas we were forcing over the water surface (to prevent any back diffusion from air by diluting the air space above the water with H₂S mixed air). In other words, the gas space above the fish was receiving the same mixed gas as the water itself, so the potential for a gradient to occur would be a minor gradient indeed.

We have added information about the spacing to the methods.

- Line 81-82: It's mentioned that fish that were in distress were removed. Were these fish put into a recovery tank and were they recovered fully? How many lost equilibrium over the 4hr period?

Only 2 fish demonstrated any equilibrium loss. They were recovered in an oxygenated water tank as per our approved SOP and returned to their tanks once recovered.

- Was O₂ of the water measured during the period of experiment? Could the surface layer have been more aerated?

For the hypoxia experiments, we did monitor the oxygen during these experiments and had ascertained that <30 minutes was required for the oxygen levels to equilibrate to the predicted mixture. Given that our system was a circulating system with constant, mixed gas aeration, this is in fact the most effective way to ensure that water is appropriately gas equilibrated. Indeed, if one were to calibrate an oxygen electrode, you would use the approach we did. However, the oxygen levels were not recorded. But we did not proceed with experiments until the electrodes read values corresponding to 2 kPa O₂. Also, prior to experiments, we verified with dye experiments that the aeration rapidly mixed the chambers, so it is not clear what to do with this comment. We did not record ASR in our hypoxia experiments. For the H₂S trials, oxygen levels are high (20.88%) in the gas we were bubbling into the water, so there is no gradient in oxygen to drive a higher surface layer oxygenation. During the hypoxia trials, the gas space had constant flow of hypoxic gas mixture, so the potential for differential oxygenation would have been minimal there.

However, we have included as supplementary a schematic of the set-up, derived from a previous publication from the lab, and from the two thesis projects that might help explain how the convective flow of water through both chambers, in combination with the constant aeration in the side chambers leads to all the right conditions for a well equilibrated and mostly uniformly mixed water system (especially after hours of bubbling). Note: we kept the entire system inside a fumehood since we could smell when an H₂S trial was being performed.

Minor comments:

- "Set point" is two words in some places and one word in others.

Thank you for spotting that error. We have changed all to consistently be two word uses.

- Figure 1B (i) and (ii) are not labelled

Thank you for noting this. It should no longer be a problem after reorganising Figure 1.

- What is the top figure in (ii) in B?

We have clarified the figure caption that the top traces are representative individuals from the bottom trace.

- Line 30: These two sentences appear to be disconnected. The previous sentence talks about the physiological role of endogenous H₂S, and this sentence lists an example of the effects of exogenous H₂S.

Thank you. R1 also noticed this. The use of the "For instance..." was misleading. We have reworded these sentences to:

"However, endogenous H₂S has roles including the physiological response to hypoxia and regulation of synaptic activity, cognitive function, inflammation, and oxygen sensing [4–6]. It has been proposed that application of exogenous H₂S in combination with low temperatures induces a drop in body temperature through entry into a hypometabolic hibernation-like state in mice [7]."

- Line 34: "... performed in small mammals *within* their thermoneutral zone,..."

Thank you. "with" has been changed to "within"

- Line 51: It's worth mentioning that H₂S inhibits the O₂ binding enzyme cytochrome oxidase in mitochondria, which also explains why the response to H₂S is similar to hypoxia.

Thank you. We had intended to make that clear, but brevity was not our friend. Inhibition of COX is indeed a principal effect of H₂S exposure, and COX sequence evolution underlies repeated colonisations of H₂S-replete habitats. We have therefore mentioned this effect and cited relevant literature.

- Line 127: "These observations point to *the* defense ..."

Thank you. We have added "the" to correct the missing article.

- Line 168: O₂ and H₂S concentrations have an inverse relationship (i.e. higher the H₂S, lower the O₂)

Sorry, we could not understand the suggestion/correction here. In the initial version, Line 168 simply stated:

"In terrestrial animals, hypoxia may be necessary for the anapyrexia effects of H₂S [7,8]."

We do not know what is being asked or suggested so cannot make the correction. We looked up and down lines in case the #168 was a typo but could not ascertain the request from the reviewer on the basis of this comment.

- Line 182: "... will depend on the *level* of environmental H₂S..."

Thank you. "degree of environmental H₂S" has been changed to "level of environmental H₂S".

- Line 182: I wouldn't view this as maladaptive for the fish (depending on how H₂S and hypoxia-tolerant they are) if it's just that cold temperatures can stimulate more H₂S to be released into the mud and change their environment.

We wished to point out some potential ecological consequences of the anapyrexia behaviour, and why it should not be assumed that 'colder is better'. Our experiments cannot at this point show adaptation, so we have rephrased the passage to read,

"Low temperatures in cold refugia can also drive redox reactions that release H₂S from mud [37], mitigating the value of the anapyrexia behavioural response. The ultimate adaptive value of the behaviour will therefore depend on the level of environmental H₂S and the combination of direct physiological and indirect impacts on microfauna and flora [3] that affect habitat suitability."

- Line 187-188: Whether it was due to the role of H₂S in O₂ sensing wasn't something that this study addressed.

We can see how this would have been interpreted as a conclusion. We have reworded this wrapping up sentence to:

“The potency of this effect might reflect its critical role in sensing and responding to oxygen levels...”

- Bibliography: check consistency in formatting of the paper titles

Thank you. We have gone through and fixed a number of journal and title inconsistencies.

Appendix B

Reviewer comments

Author Responses in Bold

Reviewer: 1

Comments to the Author(s)

The authors have addressed most of my comments in both the rebuttal as well as in the manuscript. In the process, they have greatly increased the quality of their manuscript. I only have a few additional comments.

Methods:

In a response to Reviewer 2, you mention that pilot studies were performed that indicated that H₂S levels were constant throughout the aeration period. These experiments should be noted in the methods alongside the H₂S levels measured in these experiments.

As mentioned in the previous revision, bubbling a gas mixture is how gases are calibrated, so the concern stated is unfounded and implies a misunderstanding of how aqueous gas dissolution operates. But we have now mentioned the pilot experiments in the methods:

“Pilot experiments, measuring H₂S using an H₂S electrode demonstrated constant concentrations throughout an 8 hour period of gas bubbling.”

Water pH can dramatically influence the proportion of H₂S that is present in its toxic form. What was the water pH in the shuttlebox chambers during the behavioural trials?

We previously mentioned the use of buffer in the methods and that tanks were held at pH 7.6 to 7.8. We have added further note to this effect to be clear that the buffering was maintained throughout the experiments:

“Gas dissolution equilibrated for 30 minutes, and pH was buffered within the range of 7.6 to 7.8.”

Results:

Line 171: should read “consistent” rather than “consist”

Thank you. We have changed this to “consistent”.

Discussion:

In the rebuttal, you mention that the solubility of H₂S changes quite dramatically with temperature, which may, in turn, influence the motivation of fish to seek colder environments and/or perform ASR. This caveat should be appropriately identified and addressed in the discussion.

Speculating as to whether fish detect concentration or partial pressure is a valid question (often debated in the respiratory literature) but is beyond the scope of the paper. The

change in solubility mentioned is only dramatic when comparing large temperature differences. The change in solubility would represent only a ~10% difference between 28 and 21°C, and cold water would hold higher concentrations of H₂S, so it is difficult to see how we can incorporate the reviewer's suggestion. Our initial response mentioned this, but given the 7°C change in temperature preference, the notion that zebrafish are sensing concentration vs partial pressure is something we cannot address and is not germane to the behaviour.

Line 183-185: It is unclear to me how the change in body temperature is consistent with the involvement of H₂S in O₂ sensing by NECs.

We have attempted to explain this as clearly as possible. Animals, including fish, are known to seek lower body temperatures in hypoxia. We observed that fish exposed to H₂S also seek lower body temperatures. If the signal released by hypoxic NECs is H₂S, and exogenous H₂S evokes anapyrexia, then the most parsimonious explanation is that the same pathway is involved. The alternative is to invoke as-yet undiscovered neural pathways, which would not appear to be justified at this time. We have modified the sentence as follows:

“The change in body temperature is consistent with the view that H₂S is a key effector of hypoxia sensing in fishes’ neuroepithelial cells (NECs, [5,25,26,36]), which contain H₂S-producing enzymes [26] that enhance H₂S production in response to changes in oxygen.”

Line 187-188: In the rebuttal, you give a good explanation as to why it is unclear whether hypoxia is required for the anapyrexia effects of H₂S in mammals. However, because no changes were made to the manuscript, it is still unclear to the reader why hypoxia may be necessary for the anapyrexia effects of H₂S. Providing additional details in the text would help to alleviate this issue as well as help to highlight the novelty of your data.

The argument of Hemelrijk et al. is that Blackstone et al did NOT use normoxia when they examined H₂S effects. They tested mice in 17.5% O₂, and this was necessary for the effects of H₂S. Testing H₂S in normoxia had no effect. There are also complicated time-dependent effects that they did not understand how to analyse. Hemelrijk et al conclude, “Accordingly, exogenous H₂S is a hypometabolic adjuvant rather than a hypometabolism-inducing agent.”

As for the ‘why hypoxia may be necessary for the anapyrexia effects of H₂S’, that is obviously beyond the scope of the present research. The language we used was sufficient to convey that there is uncertainty about these effects in mammals raised by other authors. By citing these studies, the reader is also pointed to a recent article challenging the original results. We feel this is sufficient for the scope of our paper that is concerned with zebrafish. We have, however, modified the sentence under discussion to the following text:

In mammals, H₂S alone or in combination with hypoxia induces anapyrexia [8,9]. Our setup precluded hypoxia, and so we can conclude that in fish, if not in mammals, H₂S induces hypometabolism rather than functioning as a hypometabolic adjuvant [9].

Minor

Throughout the text, “O₂” and “oxygen” are used interchangeably. One should be chosen and used consistently throughout the text.

We have edited the text to use oxygen where appropriate, and O₂ when referring to a level of oxygen, e.g., 2% O₂, in which case it would be unusual to write 2% oxygen.